# AG-9, an Elastin-Derived Peptide, Increases In Vitro Oral Tongue Carcinoma Cell Invasion, through an Increase in MMP-2 Secretion and MT1-MMP Expression, in a RPSA-Dependent Manner

**DOI:** 10.3390/biom11010039

**Published:** 2020-12-30

**Authors:** Clara Bretaudeau, Stéphanie Baud, Aurélie Dupont-Deshorgue, Rémi Cousin, Bertrand Brassart, Sylvie Brassart-Pasco

**Affiliations:** 1Université de Reims Champagne-Ardenne (URCA), 51100 Reims, France; clara.bretaudeau@etudiant.univ-reims.fr (C.B.); stephanie.baud@univ-reims.fr (S.B.); aurelie.dupont@univ-reims.fr (A.D.-D.); remicousin70@gmail.com (R.C.); bertrand.brassart@univ-reims.fr (B.B.); 2CNRS, UMR 7369, Matrice Extracellulaire et Dynamique Cellulaire (MEDyC), 51100 Reims, France; 3CHU Reims, Service d’Odontologie, 51100 Reims, France; 4Plateau de Modélisation Moléculaire Multi-échelle, URCA, 51100 Reims, France

**Keywords:** elastin, ribosomal protein SA, tongue carcinoma, MMP-2, EGCG

## Abstract

Oral tongue squamous cell carcinoma is one of the most prevalent head and neck cancers. During tumor progression, elastin fragments are released in the tumor microenvironment. Among them, we previously identified a nonapeptide, AG-9, that stimulates melanoma progression in vivo in a mouse melanoma model. In the present paper, we studied AG-9 effect on tongue squamous cell carcinoma invasive properties. We demonstrated that AG-9 stimulates cell invasion in vitro in a modified Boyen chamber model. It increases MMP-2 secretion, analyzed by zymography and MT1-MMP expression, studied by Western blot. The stimulatory effect was mediated through Ribosomal Protein SA (RPSA) receptor binding as demonstrated by SiRNA experiments. The green tea-derived polyphenol, (−)-epigallocatechin-3-gallate (EGCG), was previously shown to bind RPSA. Molecular docking experiments were performed to compare the preferred areas of interaction of AG-9 and EGCG with RPSA and suggested overlapping areas. This was confirmed by competition assays. EGCG abolished AG-9-induced invasion, MMP-2 secretion, and MT1-MMP expression.

## 1. Introduction

Head and neck tumors are a heterogeneous group of cancers occurring in the mouth, nose, pharynx, larynx, salivary glands, and sinuses. Approximately 90% of those cancers are squamous cell carcinomas [1]. They are typically characterized by a peak of incidence in the elderly and a strong correlation with chronic exposure to risk factors such as smoking and alcohol abuse [2]. Among head and neck squamous cell carcinomas (HNSCC), cancers of the oral cavity and pharynx display an increasing incidence since 1999, despite a decrease in tobacco use [2,3]. Many subsites have shown a rise in the number of cases, especially the anterior tongue [2]. Indeed, its incidence raised +1.8% per year on average between 2007 and 2016, particularly among young adults and especially in females [3,4,5,6,7]. Thus, in 2007, the prevalence in the US was 4422 cases, rising to 6155 cases in 2016 [3]. Despite advances in diagnosis and management of oral cancer in recent decades, the long term prognosis of patients with advanced-stage squamous cell carcinoma of the tongue is generally poor, with 5-year survival rates around 50% [8].

Elastic fibers are components of all mammalian connective tissues. Elastin, the major protein component of elastic fibers, has unique elastomeric properties which provide reversible deformability crucial for arterial vessels, lungs, and skin [9]. Elastic fibers are also stable components of tongue, providing the elasticity necessary to mechanical resistance during mastication [10].

During tumor progression, tumor and stromal cells secrete a wide variety of proteolytic enzymes including matrix metalloproteinases (MMPs), urokinase-type plasminogen activator (uPA), a disintegrin and metalloproteinases (ADAMs), a disintegrin-like and metalloproteinase with thrombospondin type 1 motif (ADAMTS), and cathepsins that are activated to degrade extracellular matrix (ECM) components and to favor tumor growth and cell dissemination. ECM-degrading enzymes also liberate active fragments from matrix components called matrikines [11]. Proteolytic degradation of elastin results in the liberation of bioactive elastin peptides, termed elastokines [12]. Elastases that degrade elastin belong to the classes of matrix metalloproteinases (MMPs), aspartic proteases, serine proteases, and cysteine proteases [13]. Elastokines containing the GxxPG motif such as VGVAPG (VG-6) promote tumor progression by stimulating cell invasion through the activation of proteolytic cascades, especially MMP-2 and uPA [14,15]. They also promote angiogenesis by stimulating the migration of endothelial cells through MMP-14 increase [16]. We have also demonstrated that an elastin nonapeptide of consensus sequence xGxPGxGxG, AGVPGLGVG (AG-9), favors the migration and invasion of tumor cells through MMP-2 and uPA increases [17,18]. These effects are mediated through the RPSA (ribosomal protein SA) receptor [19].

The 37/67-kDa laminin receptor was reported to bind elastin by Mecham et al. in 1989 [20]. The 37/67-kDa laminin receptor, RPSA, also referred as 67LR, LAMBR, LAMR1, lamR, LRP/LR, 37LRP, LBP, LBP/p40, NEM/1CHD4, SA, ICAS, and p40 is ubiquitously expressed. It allows cell adhesion to the basement membrane. The 37-, 53-, and 67 kDa forms are the major forms reported but additional high-molecular-weight (HMW) forms of 32, 37, 45, 53, 55, 67, 80, and >110 kDa were also reported. The conversion of the 37 kDa form to higher molecular weight species remains unclear [21]. RPSA not only localized on the cell surface but also in the nucleus, in association with nucleolar pre-40S ribosomes, small nucleolar ribonucleoproteins (snoRNPs), chromatin, histones, and in the cytosol as a ribosomal component or as actin and cytoskeletal stress fibers partner. It was reported to mediate cell proliferation, adhesion, differentiation, invasion, and angiogenesis. RPSA prevents cell apoptotic escape, allowing tumor progression [22].

The green tea-derived polyphenol, (−)-epigallocatechin-3-gallate (EGCG), is a small molecule with anti-cancer effects [23,24,25]. It inhibits the invasion of human oral cancer cells, decreases the production of MMPs and urokinase-plasminogen activator [26,27], prevents epithelial-mesenchymal transition [28], suppresses cell proliferation, promotes apoptosis and autophagy [29], and inhibits tumor growth [28]. Li et al. demonstrated that the inhibitory effect on cell proliferation, apoptosis, migration, and invasion of tongue squamous cell carcinoma was mediated through the hippo-TAZ signaling pathway [30]. EGCG inhibitory effect is mediated through RPSA receptor. RPSA antibodies block EGCG anti-cancer activity but do not trigger the same effects, indicating that the polyphenol may act agonistically or allosterically [21]. EGCG was reported to exert its anti-cancer activity through the 10 amino acid sequence of RPSA, IPCNNKGAHS [31].

## 2. Materials and Methods

### 2.1. Peptide Synthesis

AG-9 was purchased from Proteogenix^®^ (Schiltigheim, France). It was obtained by solid-phase synthesis using a FMOC (N-(9-fluorenyl) methoxy-carbonyl) derivative procedure. It was then purified by reverse phase high performance liquid chromatography using a C18 column, eluted by a gradient of acetonitrile in trifluoroacetic acid and lyophilized. Its purity (>98%) was assessed by HPLC and mass spectroscopy.

### 2.2. Cell Culture

Human tongue squamous cell carcinoma CAL 27 (ATCC^®^ CRL-2095^™^) were purchased from ATCC. Cells were grown in DMEM (Dulbecco’s modified Eagle medium) with 4.5 g/L glucose supplemented with 10% fetal bovine serum (FBS) at 37 °C in a humid atmosphere with 5% CO_2_ in air. At 70–90% confluency, cells were subcultured according to ATCC protocol.

### 2.3. Cytotoxicity Assay

A total of 50,000 CAL 27 cells were seeded in 96-well plates in DMEM supplemented with 10% FBS. After cell adhesion, culture medium was removed and replaced by FBS-free medium. Cells were incubated for 24 h with or without effectors. Cells were then fixed with 1.1% glutaraldehyde for 15 min and stained with crystal violet for another 15 min. Dye was eluted with a 10% acetic acid solution. Absorbance was read at 560 nm using a Biochrom Asys UVM 340 microplate reader (Biochrom, Yvelines, France).

### 2.4. Proliferation Assay

A total of 2000 CAL 27 cells were seeded in 96-well plates in DMEM supplemented with 10% FBS. After cell adhesion, medium was removed and cells were cultivated in DMEM supplemented with 2.5% FBS with or without AG-9 for 24, 48, 72, or 96 h. Cells were then fixed with 1.1% glutaraldehyde for 15 min and stained with crystal violet for another 15 min. Dye was eluted with a 10% acetic acid solution. Absorbance was read at 560 nm using a Biochrom Asys UVM 340 microplate reader (Biochrom, Yvelines, France).

### 2.5. In Vitro Invasion Assays

Invasion was assessed in modified Boyden chambers (tissue culture treated, 6.5 mm diameter, 8 µm pore, Greiner-One, Courtaboeuf, France). Further, 5 × 10^4^ cells were suspended in DMEM containing 10% FBS and seeded onto membranes coated with Matrigel^®^ (10 µg/well). After cell adhesion, culture medium was removed from the upper compartment and replaced by DMEM containing 0.2% BSA ± effectors. DMEM supplemented with 10% FBS and 2% BSA was used as a chemoattractant. After a 72 h incubation period, cells were fixed with methanol and stained with crystal violet for 15 min. Cells remaining on the upper face of the membranes were scrapped. Crystal violet was eluted using 10% acetic acid and absorbance was read at 560 nm using a Biochrom Asys UVM 340 microplate reader (Biochrom, Yvelines, France).

### 2.6. Zymography Analyses

#### 2.6.1. Cell Incubation with Effectors

At subconfluence, cells were washed twice with phosphate-buffered saline to remove residual FBS and incubated for 48 h in DMEM, with or without effectors. Conditioned media were harvested and centrifuged at 10,000× *g* for 10 min at 4 °C. 

#### 2.6.2. Gelatin Zymography

To study MMP-2 secretion by CAL 27 cells, conditioned media, diluted in 2× non-reducing Laemmli buffer, were electrophoresed in a 10% polyacrylamide SDS gel containing 0.1% (*w/v*) gelatin. The gels were washed twice for 30 min at room temperature in a 2.5% (*v/v*) Triton X-100 solution to remove SDS, then incubated at 37 °C for 24 h in 50 mM Tris-HCl (pH 7.6), 200 mM NaCl, 5 mM CaCl_2_, stained for 30 min with 0.1% (*w/v*) Coomassie blue (G 250) in 45% (*v/v*) methanol/10% (*v/v*) acetic acid and destained in the same solution without dye.

#### 2.6.3. Gelatin-Plasminogen Zymography

To study uPA secretion, CAL 27-conditioned media were electrophoresed in SDS-poly-acrylamide gels containing 1 mg/mL gelatin and 10 μg/mL plasminogen. The gels were washed twice for 30 min at room temperature in a 2.5% (*v/v*) Triton X-100 solution to remove SDS, then incubated at 37 °C for 24 h in 100 mM glycine, 10 mM EDTA (pH 8.3), stained for 30 min with 0.1% (*w/v*) Coomassie blue (G 250) in 45% (*v/v*) methanol/10% (*v/v*) acetic acid and destained in the same solution without dye.

### 2.7. Western-Blot Analyses

Samples were reduced by 10 mM of dithiothreitol and subjected to SDS-PAGE (0.1% SDS, 10% polyacrylamide gel) (50 µg total protein per lane), then transferred onto Immobilon-P membranes (Millipore, St Quentin-en-Yvelines, France). Membranes were blocked by incubation with 5% non-fat dry milk, 0.1% Tween-20 in 50 mM Tris-HCl buffer, 150 mM NaCl, pH 7.5 (TBS-T) for 2 h at room temperature. They were incubated overnight with the first antibody (anti-RPSA polyclonal antibody, Abcam Ab99484, diluted 1/3000; anti-MT1-MMP Ab38971 diluted 1/3000; anti-actin polyclonal antibody, Sigma Aldrich Biotechnology A2066, diluted 1/2000) and then for 1 h with the 1/10,000 diluted peroxidase-conjugated goat anti-rabbit secondary antibody (GE Healthcare, NA931V) at room temperature. Immune complexes were visualized using the ECL prime chemiluminescence detection kit (GE Healthcare, Orsay, France).

### 2.8. Immunocytochemistry

Cells were seeded on glass slides and incubated in 10% serum-containing medium for 16 h. Cells were incubated with an anti-RPSA antibody (Abcam, 137388) diluted 1/400 in culture medium supplemented with 1% BSA for 1 h on ice, washed and incubated for 30 min with the Alexa-488-conjugated secondary antibodies diluted 1/1000 in culture medium with 1% BSA, and then fixed for 10 min with 4% paraformaldehyde at room temperature. Immunofluorescence-labeled cell preparations were studied using a Zeiss LSM 710^®^ NLO confocal laser scanning microscope (Carl ZEISS SAS, Marly-le-Roi, France) with the 63x oil-immersion objective (ON 1.4) coupled with CHAMELEON femtosecond Titanium-Sapphire Laser (Coherent, Santa Clara, CA, USA). Alexa 488 was excited by 488 nm line of Argon. Emitted signals were collected with 493–560 nm bandpass filter. Image acquisitions were performed with ZEN Software (Carl ZEISS SAS, Marly-le-Roi, France) and all acquisition settings were constant between specimens.

### 2.9. RNA Isolation and Real-Time PCR Analysis

Total RNA was isolated from cells using a RNeasy Plus Mini kit (Qiagen, Courtaboeuf, France) according to the manufacturer’s instruction. The amount and integrity of isolated RNA were analyzed using the Bioanalyzer RNA 6000 nano assay (Agilent Technologies, Les Ulis, France) as recommended by the manufacturer. Total RNA was reverse transcribed using the first strand cDNA synthesis kit (Thermo Scientific, Illkirch, France) following the manufacturer’s instructions. Real-time PCR analysis was conducted in 20 μL reaction mixture, using Thermo Scientific Maxima SYBR Green qPCR Master Mix, following the manufacturer’s instructions. Relative expression of different gene transcripts was calculated by the ΔΔCt method. The Ct of the gene of interest was normalized to the Ct of the normalizer (EEF1A1). Fold changes (arbitrary units) were determined as 2−ΔΔCt.

RT-qPCR primers were designed according to sequence of RPSA (NM_002295). The forward primer for RPSA was 5’-CCA-TTG-AAA-ACC-CTG-CTG-AT-3’ and the reverse primer was 5’- CTG-CCT-GGA-TCT-GGT-TAG-TGA-3’ with a 144 bp product. The forward primer for EEF1a1 was 5′-CTG-GAG-CCA-AGT-GCT-AAC-ATG-CC-3′ and the reverse primer was 5′-CCG-GGT-TTG-AGA-ACA-CCA-GTC-3′ with a 221 bp product. All primers were synthesized by Eurofins (Les Ulis, France).

### 2.10. SiRNA Transfection

SiRNA were transfected as previously described [19]. SiRNA specific to human RPSA and negative control siRNA (non-targeting pool), which do not target any gene, were purchased from Qiagen. The siRNA targets different regions of the RPSA mRNA: 1st siRNA target sequence (5′-AGG-CTC-TTA-AGC-AGC-ATG-GAA-3′), 2nd siRNA target sequence (5′-TAC-CTG-GGA-TTG-CAT-ATC-AAA-3′), 3rd siRNA target sequence (5′-TTG-CAT-ATC-AAA-GCA-TAA-TAA-3′), and 4th siRNA target sequence (5′-TCG-ACA-TGA-GTT-GTA-CTT-CTA-3′). Expression of RPSA mRNA and protein was confirmed by real-time PCR and Western blot.

### 2.11. Docking Experiments

Dockings of the AG-9 peptide and EGCG onto RPSA (RCSB Protein Data Bank 3BCH) were performed using Autodock software (version 4.2) [32]. We performed preliminary docking experiments to determine the relevant set of docking parameters. The software was used with a fixed RPSA and flexible AG-9 and EGCG ligands. Since RPSA is a large molecule, we performed several independent dockings targeting subvolumes of the protein; we considered 80 overlapping boxes with a volume of 31.5 Å × 31.5 Å × 31.5 Å. The spacing parameter used to compute the 3D maps in each box was set to 0.25 Å. The selected search method was the Lamarckian genetic algorithm, and for each docking experiment, 50 solutions per box were generated with the default parameters of Autodock except for the population size (200), number of energy evaluations (2.5 × 10^6^), and maximum number of generations (270,000), which were derived from the preliminary study. The identification of the contacts between RPSA and the poses of each ligand was carried out by the evaluation of the intermolecular atomic distances. Contacts were counted for distances lower than the 5 Å threshold. Molecular models were graphed with VMD software, which is available online.

### 2.12. Statistical Analysis

Results were expressed as means +/− standard deviation. Statistical significance between groups was assessed using unpaired Student’s *t* test.

## 3. Results

### 3.1. CAL 27 Cells Express RPSA Receptor

Elastin was reported to interact with invasive cancer cells through RPSA [33]. Previous experiments from our laboratory identified RPSA as AG-9 receptor on HT-1080 human fibrosarcoma cell surface [19], and on MIA PaCa-2 cell pancreatic adenocarcinoma cells [34]. We first checked for RPSA expression in CAL 27 cells, compared to MIA PaCa-2 cells. By qPCR and Western blot analyses, we proved that RPSA was expressed by CAL 27 at the mRNA and protein level (Figure 1A,B respectively). By immuno-histochemistry, we confirmed that RPSA was present at the cell surface of CAL 27 cells.

### 3.2. AG-9 Increases CAL 27 Invasion through Matrigel^®^, MMP-2 Secretion, and MT1-MMP Expression

Soluble kappa-elastin peptides were shown to regulate MT1-MMP and MMP-2 [14]. Both MMPs are largely involved in cell invasion. Recently, we reported that AG-9 peptide stimulates tumor cell invasion at lower concentrations than the well characterized VG-6 peptide [18]. CAL 27 invasion was studied in vitro in the presence of soluble kappa-elastin peptides or AG-9 peptide using the transwell invasion assay. After 72 h of incubation, kappa-elastin peptides and AG-9 peptide increased cell invasion by +22 and +25%, respectively (Figure 2A). The increase was independent of cell proliferation since the peptides did not significantly modified cell proliferation as demonstrated using the crystal violet assay (Figure 2B). MMP-2 secretion was studied by zymography. After 24 h of incubation, kappa-elastin peptides and AG-9 peptide increased MMP-2 cell secretion by +39 and +139%, respectively (Figure 2C). Kappa-elastin peptides and AG-9 peptide also induced MT1-MMP expression, studied by Western blot (Figure 2D).

### 3.3. AG-9 Proinvasive Effects Are Mediated through RPSA Receptor

CAL 27 cells were tranfected with control (non-targeting) or RPSA siRNA. RPSA gene expression was measured by qPCR 48 h after transfection. RPSA gene expression was decreased by 62% after RPSA siRNA transfection compared to control (Figure 3A). RPSA protein expression was evaluated 72 h after transfection by Western blot. RPSA was decreased by 36% (Figure 3B). We performed invasion assays, zymography, and Western blot experiments. Even partial RPSA invalidation abolished the AG-9-induced effects on MMP-2 secretion (Figure 3C) and MT1-MMP expression (Figure 3D). The results confirm the involvement of RPSA receptor in AG-9-mediated effects.

### 3.4. In Silico Study of AG-9 and EGCG Binding on RPSA

EGCG was previously reported to bind RPSA [18,22,35]. EGCG may prevent AG-9 fixation on RPSA and represent a molecule of choice to limit AG-9 pro-tumoral effects on oral tongue squamous cell carcinoma. Clustering the 50 best results for each ligand, we first identified the preferred areas of interaction (PAI) with EGCG and then with AG-9 and found overlapping areas (Figure 4A,B). The comparison of the associated distribution profile of the free energy of binding demonstrated lower free energy of binding for EGCG onto RPSA than for AG-9 onto RPSA, corresponding to a stronger binding of the polyphenol onto the receptor (Figure 4C,D). The localization of the lowest free energy of binding pose of each ligand highlighted a colocalization onto RPSA (Figure 4E,F). Finally, the analysis of the RPSA residues making contact with the ligands evidenced common interactions with R^117^, ^120^RL^121^, and the region ^140^VNLP^143^ (Figure 3G).

### 3.5. In Vitro EGCG Cytotoxicity on CAL 27 Cells

The aim of this part was to determine EGCG maximal concentration that could be used to counteract AG-9 stimulation without affecting cell viability. For this purpose, CAL 27 cells were incubated for 24 h with increasing amount of EGCG and cell viability was measured (Figure 5). Cell viability was 97.4% for 10 µM EGCG. Comparable results were obtained by Weisburgh et al. [36]. The same authors reported that cell viability of normal oral fibroblasts was also around 95% at this concentration.

We thus decided to use this concentration in the following experiments to try to block AG-9-stimulatory effects without affecting cell viability.

### 3.6. EGCG Prevents AG-9 Stimulatory Effect on CAL 27 Migration, Invasion, MMP-2 Secretion, and MT1-MMP Expression

Incubation with 10µM EGCG slightly decreased CAL 27 migration (−11%; Figure 6A) and significantly decreased cell invasion (−23%; Figure 6B) and it abolished AG-9 stimulatory effect (Figure 6). It also slightly decreased MMP-2 secretion and MT1-MMP expression (Figure 6C,D). This is in accordance with previously published papers [37]. Incubation with EGCG also prevented AG-9 stimulatory effect on cell migration (Figure 6A), invasion (Figure 6B), as well as on MMP-2 secretion (Figure 6C) and MT1-MMP expression (Figure 6D).

## 4. Discussion

Oral squamous cell carcinoma (OSCC) is the most common malignant epithelial neoplasm affecting the oral cavity. The treatment of choice for OSCC is surgical resection. Adjuvant radiotherapy with or without chemotherapy is offered when there is a high risk of recurrence and after taking into consideration multiple factors, including patient’s age and comorbidities, pathologic staging, margin status, the extent of nodal involvement, and other histopathologic characteristics of the primary tumor [38]. Despite the advances of therapeutic approaches, percentages of morbidity and mortality of OSCC have not improved significantly during the last 30 years. Developing new therapeutic approaches is thus challenging.

Cancer development leads to ECM degradation by tumor and stromal cells. Fragments with biological activities are released during this process, named matrikines. Elastin is the major component of elastic fibers. Its cleavage by elastase-proteinases such as metalloproteinases or leucocyte elastase is known to unmask cryptic sites within the macromolecule and to release matrikines, called elastin derived peptides (EDPs) or elastokines. These EDPs exert a wide range of biological activities. They influence cell survival, differentiation [39], proliferation, chemotaxis [18,40], migration [18,41], tumor progression [15,17,19,42], angiogenesis [16], atherogenesis, and aneurysm formation. Among all the EDPs described in the literature, two major consensus sequence were reported: the xGxxPG consensus sequence including the VGVAPG, VAPG, VGVPG, VGAPG, (VGVAPG)_n_, and PGAIPG peptides and the xGxPGxGxG consensus sequence with the AGVPGLGVG, AGVPGFGVG, GLGVGVAPG, and GFGVGAGVP peptides. In vivo study showed that AG-9 peptide promotes melanoma progression even more than the well described VG-6 peptide. These results were confirmed by in vitro studies in proliferation assays, migration assays, adhesion assays, proteinase secretion studies, and pseudotube formation assays to investigate angiogenesis [18,34].

In the present paper, we report for the first time EDP-effect on oral tongue SCC. CAL 27 is one of the most frequently used cell lines in the field of OSCC studying [43]. Soluble elastin peptides were obtained by partial hydrolysis of elastin in 1 M KOH in 80 per cent aqueous ethanol (kappa-elastin). Additionally, 50 mM kappa-elastin increases CAL 27 invasion through transwell previously coated with Matrigel^®^ which mimicks basement membrane. AG-9 also significantly increased CAL 27 invasion, as previously reported for MIA PaCa-2 pancreatic ductal adenocarcinoma cells [34]. EDP were previously reported to increase MMP secretion, especially MMP-2 [44,45]. These results were confirmed by zymography analysis of conditioned media from CAL 27. AG-9 increased MMP-2 secretion, as reported for MIA PaCa-2 cells; this effect was biphasic, with an optimal effect obtained for 1.10^−7^ M AG-9. Moreover, AG-9 increases MT1-MMP expression. MT1-MMP is able to degrade extracellular matrix by itself or to activate proMMP-2 at the cell surface.

The pro-tumor biological effects of AG-9 was previously reported to involve a lactose-insensitive receptor, the ribosomal protein SA (RPSA) [19]. The 37/67-kDa laminin receptor was reported to bind elastin by Mecham et al. in 1989 [20]. It allows cell adhesion to basement membrane. RPSA does not only localize on the cell surface but also in the nucleus, in association with nucleolar pre-40S ribosomes, small nucleolar ribonucleoproteins (snoRNPs), chromatin, histones, and in the cytosol as a ribosomal component or as actin and cytoskeletal stress fibers partner. It was reported to mediate cell proliferation, adhesion, differentiation, invasion, and angiogenesis. RPSA also prevents cell apoptotic escape, allowing tumor progression [22].

The green tea-derived polyphenol (−)-epigallocatechin-3-gallate (EGCG), was reported to affect cell behavior through RPSA binding [31]. We performed molecular docking experiments to determine its potential preferred areas of interaction with RPSA and especially the area with the lowest energy of binding that means the highest probability of binding. We did the same for the AG-9 peptide. The preferred area of interaction with the lowest energy for EGCG and AG-9 was broadly overlapping. We then formulated the hypothesis that EGCG could counteract the effects induced by the peptide AG-9. To test this hypothesis, we performed the invasion experiments in presence or absence of EGCG. We first observed that EGCG alone at 10 µM was able to decrease CAL 27 invasion. This was previously reported by Chang et al. with a concentration of 25 µM EGCG [37]. EGCG was also able to inhibit SCC-4, SCC-9, and SCC-15 cell [28,29,30]. other oral SCC and nasopharyngeal carcinoma cell invasion [46,47]. In addition, 10 µM EGCG also inhibits MMP-2 secretion as previously described for CAL 27 cells [37], nasopharyngeal carcinoma [47], and buccal mucosa cancer cells [26]. This was also observed in other oral SCC. Further, 10 µM EGCG prevents AG-9 induced invasion, MMP-2 secretion, and MT1-MMP expression. As suggested by molecular docking experiments, this may be due to steric hindrance or conformation modification as the PAIs of lowest binding energy of both molecules are very close. Binding energy of EGCG for RPSA are lower than those found with AG-9, suggesting that RPSA affinity for EGCG may be greater than for AG-9. 

## 5. Conclusions

During cancer progression, tumor cells induce the expression of MMPs, such as MT1-MMP and MMP-2, which degrade ECM macromolecules like elastin and release EDPs, such as AG-9, that in turns stimulate MMP secretion, leading to an auto-amplification loop. EGCG prevents AG-9 stimulation and represents a molecule of choice to limit cancer progression in elastin-rich tissues.

## Figures and Tables

**Figure 1 biomolecules-11-00039-f001:**
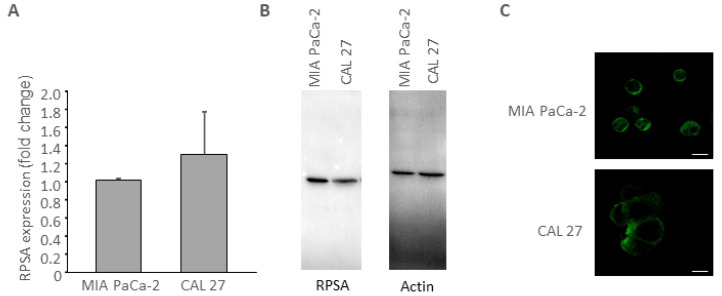
RPSA expression was studied at the mRNA level by qPCR (**A**) and at the protein level by Western blot (**B**). RPSA distribution at the cell surface was studied by immunocytochemistry (orthoslide) (**C**). Scale bar: 20 µM.

**Figure 2 biomolecules-11-00039-f002:**
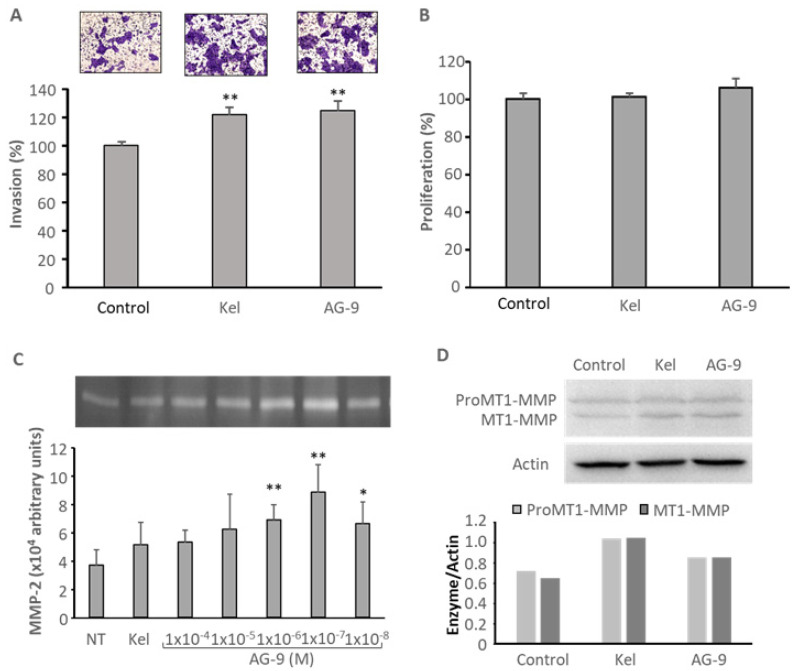
Cell invasion was studied using the transwell assay system coated with Matrigel^®^ as described in the Materials and Methods section after 72 h of incubation with the different effectors: Kappa-elastin peptides 50 µg/mL, AG-9 peptide 10^−7^ M. **: *p* < 0.01 (**A**). Cell proliferation was assessed using crystal violet staining after 72 h of incubation with the effectors: kappa-elastin peptides 50 µg/mL, AG-9 peptide 10^−7^ M (**B**). MMP-2 secretion was studied by zymography after 24 h of incubation with kappa-elastin peptides 50 µg/mL or various amount of AG-9 ranging from 10^−4^ M to 10^−8^ M. *: *p* < 0.05; **: *p* < 0.01 (**C**). MT1-MMP expression was studied by Western blot after 24 h of incubation with the effectors: kappa-elastin peptides 50 µGg/mL, AG-9 peptide 10^−7^ M (**D**).

**Figure 3 biomolecules-11-00039-f003:**
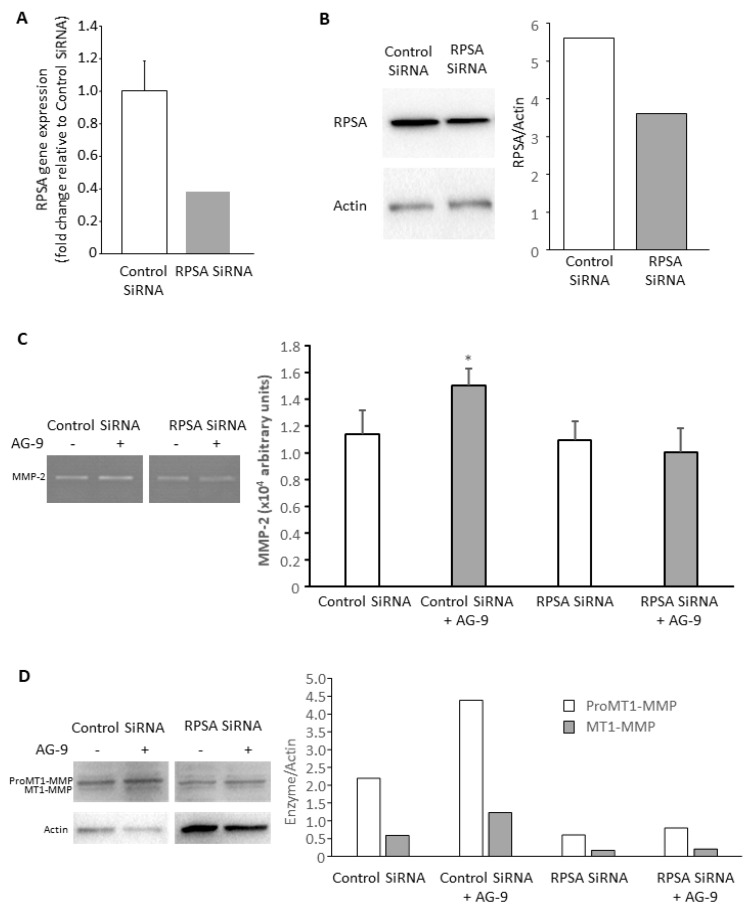
Cells were transfected with control or RPSA siRNA. RPSA gene expression was measured 48 h post-transfection (**A**) and protein expression was measured 72 h post-transfection (**B**). For MMP-2 secretion (**C**) and MT1-MMP expression (**D**) studies, cells were incubated for another 24 h with or without 10^−7^ M AG-9 peptide. *: *p* < 0.05.

**Figure 4 biomolecules-11-00039-f004:**
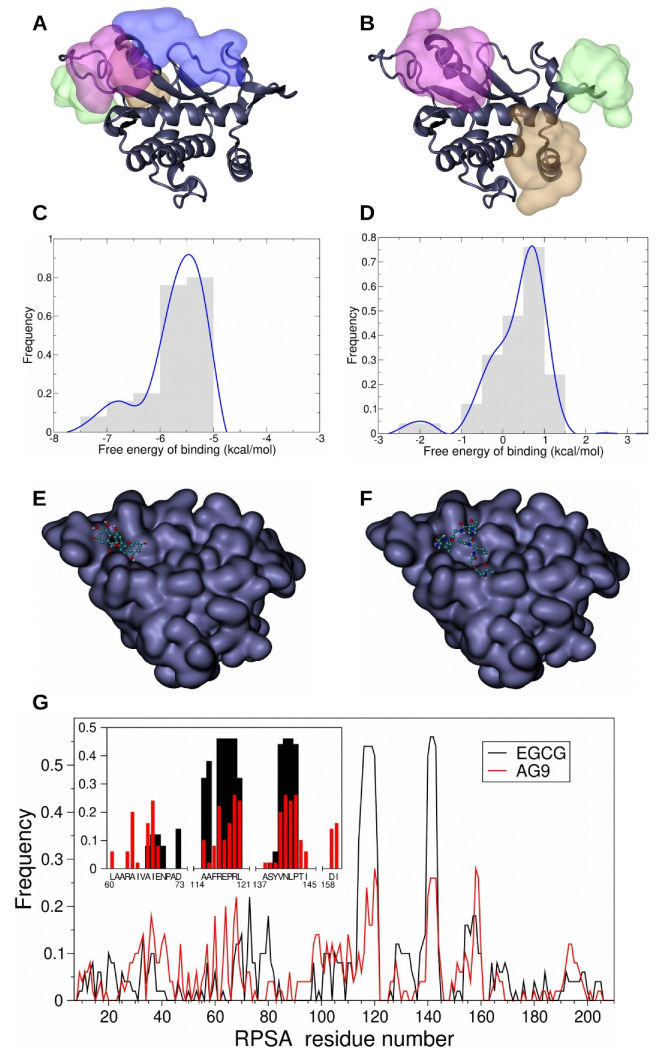
Docking experiments of green tea-derived polyphenol EGCG and AG-9 peptide onto RPSA were performed using Autodock software and evidenced the existence of 4 preferred area of interaction (PAI) for EGCG (**A**) and 3 PAI for AG-9 (**B**). The color code used for the representation of PAI is linked to their population: the most populated one is represented in pink, the second in green, the third in brown, and the fourth in blue. Comparison of the 50 best results of EGCG and AG-9 docking experiments displays an overlapping area (pink PAI). Frequence distribution diagrams of the free energy of binding of the 50 best poses of EGCG onto RPSA demonstrated that those free energies of binding are comprised between −7.75 and −4.75 kcal/mol. The most frequent poses have a free energy of binding of −5.5 kcal/mol (**C**). For AG-9 peptide, the free energy of binding of the 50 best results of docking onto RPSA is comprised between −2.75 and +1.75 kcal/mol and the energy of binding of the most frequent position of AG-9 onto RPSA is around 0.75 kcal/mol (**D**). The best docking poses of EGCG (**E**) and AG-9 (**F**) are located on the same area at the surface of RPSA. Comparison of the frequency of contacts made by RPSA residues with the 50 best results of EGCG (black line) and AG-9 (red line) reveals two favored regions (frequency above the threshold of 0.2) of the protein (**G**). A focus on the poses associated with the first cluster of EGCG and AG-9 (G/close-up) allows the identification of the following hot spots (frequency above the threshold of 0.2 for the two ligands): R^117^, ^120^RL^121^, and the sequence ^140^VNLP^143^.

**Figure 5 biomolecules-11-00039-f005:**
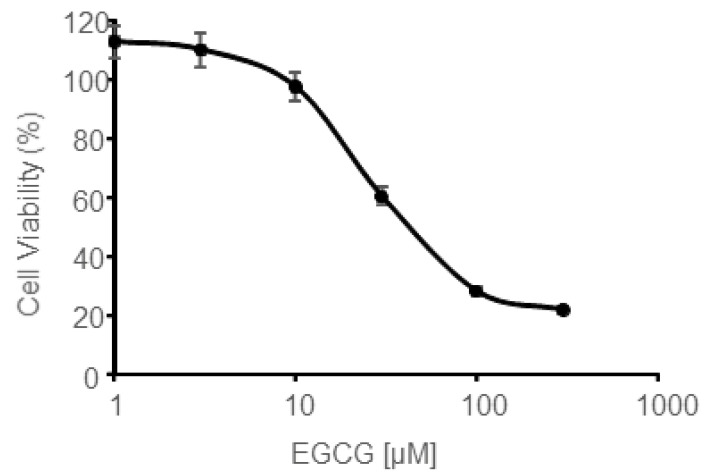
Cell viability was assessed using the crystal violet assay after 24 h of incubation with increasing concentrations of EGCG ranging from 1 to 300 µM.

**Figure 6 biomolecules-11-00039-f006:**
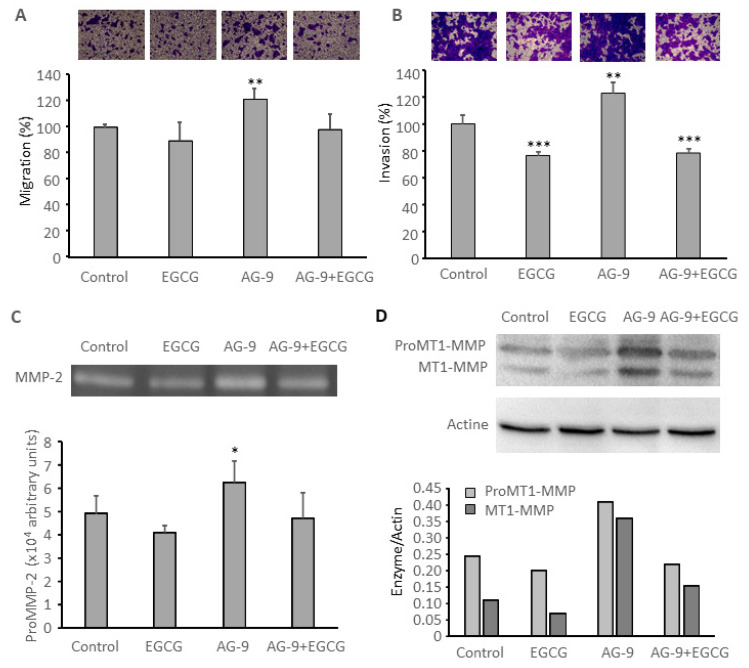
Cell migration was studied using the transwell assay system after 48 h of incubation with the different effectors: AG-9 10^−7^ M and EGCG 10 µM. **: *p* < 0.01 (**A**). Cell invasion was studied in transwells previously coated with Matrigel^®^. ***: *p* < 0.001, **: *p* < 0.01 (**B**). MMP-2 secretion was studied by zymography after 24 h of incubation with the different effectors: with the different effectors: kappa-elastin peptides 50 µg/mL, AG-9 peptide 10^−7^ M. *: *p* < 0.05 (**C**). MT1-MMP expression was studied by Western blot after 24 h of incubation with the different effectors: kappa-elastin peptides 50 µg/mL, AG-9 peptide 10^−7^ M (**D**).

## Data Availability

Data is contained within the article.

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
