# Peer review of "AG-9, an Elastin-Derived Peptide, Increases In Vitro Oral Tongue Carcinoma Cell Invasion, through an Increase in MMP-2 Secretion and MT1-MMP Expression, in a RPSA-Dependent Manner"

_biomolecules, 2020, doi:10.3390/biom11010039_

Round 1
Reviewer 1 Report
1. Please check typo errors in the manuscript
2. Use normal cells to carry out cytotoxic evaluations
3. Improve gel images (supply full membranes -uncropped gel images to the journal)
4. Please perform cell migration assay and colony formation assay with the peptide and EGCG
The authors have evaluated the cancer (tongue) inhibitory effects of EGCG induced by an elastin derived peptide
Although the manuscript is interesting, i see some parts to be changed to improve the overall quality of the work.
1. Please change the title and put a new title in a meaningful way
2. Also improve the abstarct
3. Add more information about the anticancer effects of EGCG
4. Discuss tongue cancer statistics in the introduction with latest cancer statistics
Thanks
Author Response
Dear Reviewer,
We thank you for your careful review and valuable comments which helped us to improve the quality of our manuscript.
We took into account your remarks and wrote a point-by-point response (Please see the attachment). We highlighted modifications and corrections using the "Track Changes" function in Microsoft Word in this revised manuscript.
Best regards
Dr Sylvie BRASSART-PASCO

Reviewer 2 Report
Bretaudeau and coauthors have performed a series of experiments to investigate how elastin peptide product AG-9 stimulates human tongue squamous cell carcinoma in situ. They have proposed that the laminin receptor RPSA is responsible for mediating this response and that the polyphenol EGCG can act as an antagonist for AG-9. While they were able to show that RPSA is present in their CAL7 cells (Western, qPCR and immunohistochemically), that AG-9 can stimulate tumor invasion phenotype (Matrigel invasion), and that EGCG could abrogate this invasion, they did not actually demonstrate that RPSA mediates this response. They did provide a docking study that suggests that AG-9 and EGCG colocalize on the RPSA to the same binding site. Prior studies have supported that RPSA is bound by both AG-9 and EGCG and the authors allude to studies that suggested that RPSA was mechanistically identified to the receptor in other cell lines, but this work does not actually demonstrate that the invasion stimulatory phenotype of AG-9 upon CAL7 is mediated through RPSA. No direct biophysical characterization of AG-9/EGCG antagonism was investigated and certainly not with CAL7 surface specific isoform(s). In the absence of direct demonstration that RPSA mediates the response, it is hard to confirm the implied mechanistic role and suggests the following question: Does RNAi knockdown of RPSA eliminate the CAL7 cells response to AG-9? That RNAi experiment is missing and would elevate this work tremendously. In the absence of that experiment the docking is disconnected from cell studies and seems like the authors are reaching for a model that is not yet demonstrated (likely to be true, but not yet demonstrated).
Recommendation: Perform partial or complete RNAi of RPSA and assess if the knockdown eliminates response to AG-9. If it does not, then another receptor may be involved. If complete knockdown of RPSA induces apoptosis (possible, see Vania et al 2020 BMC cancer), then partial knock down may be achievable to probe for partial reduction in AG-9 response. Alternatively, some form of cross-linked MS/MS or FRET studies may be able to identify if AG-9 is actually binding to and mediating through RPSA in live CAL7 cells.
Author Response
Thanks for your valuable comments which helped us to improve the quality of our manuscript.
We took into account your remarks and wrote a point-by-point response (Please see the attachment). We highlighted modifications and corrections using the "Track Changes" function in
Microsoft Word in this revised manuscript.
Best regards
Dr Sylvie Brassart-Pasco.

Reviewer 3 Report
Previously, the authors have reported that an elastin-derived nonapeptide peptide, AG-9, stimulated melanoma progression in vivo in a mouse melanoma. In this study, they examined the effect of AG-9 on tongue squamous cell carcinoma. AG-9 enhanced invation of human Tongue Squamous Cell Carcinoma CAL 27 invasion through Matrigel by increasing the expression of MT1-MMP and MMP-2. AG-9 is known to the lganda of RPSA (Ribosomal Protein SA) receptor. (−)-epigallocatechin-3-gallate (EGCG),is the most abundant and the most biologically active catechin in green tea. EGCG has been shown to have protective effects on oxidative damage and anti-inflammation. EGCG is also know to affect cell behavior through RPSA binding. Thus, the authors have investigated the affect of EGCG on CAL 27 cells. Viability of CAL27 cells was reduced as the increase of EGCG concentration. EGCG prevented AG-9 stimulatory effect on CAL 27 invasion and MMP-2 secretion. Molecular docking simulation shows that EGCG shares with the same binding site of AG-9. Thus, it is reasonable to think that EGCG prevents the binding of AG-9.
The experiments were well designed and the results are all reasonable for the conclusion. Therefore, I recommend this manuscript for publication in Biomolecules.
Author Response
We thank you for your review of our manuscript.
The manuscript was carefully reread and checked for errors.
We highlighted modifications and corrections using the "Track Changes" function in Microsoft Word in this revised manuscript.
Best regards,
Dr Sylvie Brassart-Pasco.
Round 2
Reviewer 1 Report
Accept
Reviewer 2 Report
The authors added SiRNA knockdown studies that connect Ag-9 action to RPSA, but have not probed if the prior suggestion that EGCG can disrupt via RPSA. That is to say their experiment that showed disruption of invasion in the presence of EGCG was not abolished in SiRNA transfected cells. However, they authors softened their conclusions (relating to EGCG involvement) in the title and document to reflect this so I suppose that is ok. No further changes are required IMO.